# Peripancreatic Adipose Tissue Remodeling and Inflammation during High Fat Intake of Palm Oils or Lard in Rats

**DOI:** 10.3390/nu13041134

**Published:** 2021-03-30

**Authors:** Jonas Laget, Youzan Ferdinand Djohan, Laura Jeanson, Karen Muyor, Eric Badia, Jean Paul Cristol, Charles Coudray, Christine Feillet-Coudray, Claire Vigor, Camille Oger, Jean-Marie Galano, Thierry Durand, Anne-Dominique Lajoix, Nathalie Gayrard, Bernard Jover

**Affiliations:** 1RD-Néphrologie, 34090 Montpellier, France; karenmuyor@gmail.com (K.M.); nathalie.gayrard@umontpellier.fr (N.G.); 2BC2M, UR-UM 101, UFR Pharmacie, Université de Montpellier, 34090 Montpellier, France; laura.jeanson@inserm.fr (L.J.); anne-dominique.lajoix@umontpellier.fr (A.-D.L.); 3PHYMEDEXP, INSERM, CNRS, Université de Montpellier, 34090 Montpellier, France; djohanferdinand@gmail.com (Y.F.D.); eric.badia@umontpellier.fr (E.B.); jp-cristol@chu-montpellier.fr (J.P.C.); bernard.jover@inserm.fr (B.J.); 4DMEM, EMN, UMR 866, INRAe, Université de Montpellier, 34090 Montpellier, France; charles.coudray@inrae.fr (C.C.); christine.coudray@inrae.fr (C.F.-C.); 5IBMM, UMR 5247, CNRS, Université de Montpellier, ENSCM, 34090 Montpellier, France; claire.vigor@umontpellier.fr (C.V.); camille.oger@umontpellier.fr (C.O.); jean-marie.galano@umontpellier.fr (J.-M.G.); thierry.durand@umontpellier.fr (T.D.)

**Keywords:** palm oil, high fat intake, adipose tissue, inflammation, lipid oxidation

## Abstract

Excessive fat consumption leads to the development of ectopic adipose tissues, affecting the organs they surround. Peripancreatic adipose tissue is implicated in glucose homeostasis regulation and can be impaired in obesity. High palm oil consumption’s effects on health are still debated. We hypothesised that crude and refined palm oil high-fat feeding may have contrasting effects on peripancreatic adipocyte hypertrophy, inflammation and lipid oxidation compound production in obese rats. In Wistar rats, morphological changes, inflammation and isoprostanoid production following oxidative stress were assessed in peripancreatic adipose tissue after 12 weeks of diets enriched in crude or refined palm oil or lard (56% energy from fat in each case) versus a standard chow diet (11% energy from fat). Epididymal white and periaortic brown adipose tissues were also included in the study. A refined palm oil diet disturbed glucose homeostasis and promoted lipid deposition in periaortic locations, as well as adipocyte hypertrophy, macrophage infiltration and isoprostanoid (5-F_2c_-isoprostane and 7*(RS)*-ST-Δ^8^-11-dihomo-isofuran) production in peripancreatic adipose tissue. Crude palm oil induced a lower impact on adipose deposits than its refined form and lard. Our results show that the antioxidant composition of crude palm oil may have a protective effect on ectopic adipose tissues under the condition of excessive fat intake.

## 1. Introduction

High fat intake, associated with a sedentary lifestyle, contributes to the development of obesity and obesity-related disorders. Yet, the type of fatty acids provided may result in different levels of metabolic alterations. Palm oil is available in different forms, which include crude or red palm oil (crude PO) and refined palm olein (refined PO) [1]. Despite a known beneficial influence of crude PO in cardiovascular disease and its many nutritional qualities and benefits [2,3], the presence of palmitic acid in palm oil is an argument to avoid palm oil consumption (see [4] for a review), on top of questions about the sustainability of its production [5]. However, crude PO contains compounds such as carotenoids and tocopherols which may provide health benefits [1,6,7]. Yet, the evaluation of PO effects on health has mainly focused on blood lipids, cardiovascular diseases, cancer and type 2 diabetes, with mixed results [8,9].

When over-consumed, a fat-rich diet leads to lipid accumulation in various adipose tissues (ATs). No longer regarded as a storage depot only, fat is an endocrine organ producing adipokines that act locally and systemically to modulate energy homeostasis, insulin resistance and inflammatory status [10,11]. Specific structural and functional changes are observed in obese visceral adipose tissue, together with local inflammation and adipokine production, promoting metabolic disturbances [12,13]. Obesity leads to the presence of ectopic fat surrounding organs and to the accumulation of lipids in tissues themselves. In the liver, short-term feeding with an enriched palm oil diet was associated with a more marked hepatic lipid accumulation compared to a sunflower oil-enriched diet. It was concluded that palm oil further altered liver lipid metabolism [14]. However, we previously reported that excessive and long-term intake of crude or refined PO had no significant effect on liver steatosis, while it did affect glucose tolerance [15]. Adipose tissue’s association with various organs suggests local interactions. In the heart, epicardial adipose tissue may increase pro-inflammatory cytokine production, such as IL-6, which affects local endothelium [16]. Recently, Al-Dibouni et al. [17] showed that both TNF-α and IL-6 mRNA levels were enhanced specifically in pericardial adipose tissue in mice. On the other hand, local biocommunication between the pancreas and the surrounding adipose tissue occurs and is altered during obesity [18,19]. Secreted factors from peripancreatic white adipose tissue (pWAT) of Zucker diabetic fatty (ZDF) rats were reported to impair insulin secretion, cellular proliferation and apoptosis of INS-1 cells [20]. Moreover, in insulin-resistant patients, pancreatic fat content could favour a local pro-inflammatory environment and was negatively associated with insulin secretion [21]. Adipose tissue remodelling and inflammatory status following high fat intake are dependent on the lipid [22] and antioxidant [23,24] composition of the diets. In this context, specific local pro-inflammatory changes in pWAT after PO-rich diets may have systemic consequences regarding insulin resistance and glucose homeostasis.

On a different note, high reactive oxygen species (ROS) production can appear in AT during obesity and AT accumulation correlates with an increase in systemic oxidative stress [25]. Oxidative stress regulation is of major concern for adipose tissue metabolism since it influences its inflammation [26] and can protect it from insulin resistance [27]. Free radical production in AT may lead to an increase in local isoprostanoid formation and release. Isoprostanoids are a family of oxidised lipids produced by the non-enzymatic oxidation of fatty acids, mainly arachidonic acid, from membrane phospholipids. Since the discovery of isoprostanoid formation in vivo [28], they have been widely studied to assess circulating or tissue oxidative stress in numerous pathologies [29]. Moreover, isoprostanoids are bioactive compounds that can influence many processes such as angiogenesis, macrophage migration and inflammation, thus participating in adipose tissue physiology [30,31].

In the present study, we hypothesised that high-fat feeding induces ectopic lipid deposition in peripancreatic adipose tissue, affecting inflammation and lipid oxidation compound production locally. The effect of refined PO was compared to crude PO and to a diet enriched with lard. Our results indicate that unlike crude PO, a refined PO diet induces specific changes in pWAT in terms of adipocyte hypertrophy, inflammation and oxidative stress.

## 2. Materials and Methods

### 2.1. Animals and Diets

Twenty-eight male Wistar rats (Charles River, L’Arbresle, France) aged 6 weeks at the beginning of experiments were housed under conditions of constant temperature (20–22 °C), humidity (45–50%) and a standard dark cycle (20.00–08.00 h) with free access to food and water. Rats were assigned to four groups of seven animals and fed for 12 weeks either a standard rat chow diet (control group) or one of the three high-fat (HF) diets. In the control diet, 11% of the energy was given by fat (5% soybean oil) whereas in enriched diets, 56% of the energy was provided by fat intake [15]. The fat-enriched diets consisted in 2.5% (*w/w*) of soybean oil and 30% (*w/w*) of crude PO, refined PO (*Elaeis guineensis tenera*, Sanya Cie, Abidjan, Côte d’Ivoire) or lard (Alva, Rezé, France). The fatty acid composition of oils is reported in Appendix A. Typical amounts of vitamin E, carotenoids and polyphenols in the crude and refined PO are shown in Appendix A. The experiments complied with the guidelines for the care and use of laboratory animals (National Academies Press US, 8th edition, 2011) and all procedures were approved by the local ethical committee (reference CEEA-LR-12002, Montpellier, France).

### 2.2. Tissue Sampling and Blood Parameters

Rats were fasted overnight, and blood was sampled via the abdominal artery under pentobarbital anaesthesia (55 mg/kg ip). After centrifugation at 1000× *g* for 10 min at 4 °C, plasma was collected and stored at −80 °C for glucose and insulin measurement.

Then, epididymal (eWAT), periaortic brown (AoBAT) and peripancreatic adipose tissue (pWAT) were removed, rinsed in PBS and flash frozen in liquid nitrogen. One piece of each was also fixed in 10% formalin before paraffin embedding for histological studies.

Plasma glucose, blood triglycerides and cholesterol were determined on a COBAS automated analyser (Roche Diagnostics, France) and insulin was quantified by an immunoassay kit (Mercodia Rat Insulin ELISA). The homeostasis model assessment of insulin resistance (HOMA-IR) was calculated using the follow equation: HOMA-IR = (fasting blood glucose in mM x fasting plasma insulin in mUI/L)/22.5.

### 2.3. Isoprostanoid Quantification

Isoprostanoids were extracted from peripancreatic adipose tissue according to the protocol described by Lee et al. [32]. Briefly, 200 mg of frozen samples were homogenised using Fast-Prep 24 (MP Biomedical, Santa Ana, CA, USA) in the presence of butylated hydroxytoluene (BHT), methanol and ethylene glycol-bis(2-aminoethyl- ether)-N,N,N′,N′-tetraacetic acid (EGTA). After the addition of internal standards into samples, the lipid fraction was recovered with Folch extraction. Then, alkaline hydrolysis was performed and total isoprostanoids were included in the analysis (esterified and free forms). They were purified using solid phase extraction (SPE) anion exchange cartridge separation (Waters, Milford, MA, USA) by a succession of washings and finally eluted in a mixture of hexane, ethanol and acetic acid. The elution solvent was evaporated under a nitrogen stream and concentrated residue was then solubilised. Isoprostanoid concentration was determined by liquid chromatography (LC)–mass spectrometry (MS), as recently described [33]. Note that for 5(*RS*)-5-F_2t_-IsoPs, 5(*RS*)-5-F_2c_-IsoPs and 7*(RS)*-ST-Δ^8^-11-dihomo-isofurans, the assay could not differentiate between the R/S isomers. Whatever the conditions used, each dosage was performed in duplicate.

Isoprostanoid concentrations were obtained in picograms (pg) of isoprostanoids per gram of adipose tissue and we chose to convert this value to pg per adipocyte. The average surface (cross-sectional area in µm^2^) of the adipocytes for each histological sample was determined as described in the next section. Assuming that fat cells were spherical, their average radius was calculated and we deduced their volume in cm^3^. Then, to determine adipocyte number per gram of TA, we considered an approximate density of pWAT of 0.94 (based on measurements of eWAT in the study of Rotondo et al. [34]), so 1 g of pWAT represented 1.064 cm^3^ ($=\frac{1}{0.94}$). The calculations steps were realised as follows:$$\mathrm{Adipocyte}\;\mathrm{radius}\;=\;\sqrt{\frac{area}{ᴨ}},$$
$$\mathrm{Adipocyte}\;\mathrm{volume}\;=\;\frac{4}{3}\;\times \;ᴨ\;\times \;radius{\;}^{3},$$
$$\mathrm{Adipocytes}\;\mathrm{number}\;\mathrm{per}\;\mathrm{gram}\;\mathrm{of}\;\mathrm{TA}\;=\;\frac{Volume\;of\;1\;g\;of\;TA}{Adipocyte\;Volume}\;=\;\frac{\;1.064}{Adipocyte\;Volume\;\left(cm\;3\right)}$$

Isoprostanoid amount per adipocyte was deduced and expressed as femtograms (fg) of isoprostanoids per adipocyte, for convenience.

### 2.4. Immunohistochemistry and Haematoxylin Staining

Adipose tissue samples were fixed in 10% formalin solution (Sigma-Aldrich, Saint-Quentin Fallavier, France) for at least 24 h, and embedded in paraffin. Sections of 5 µm were cut on a Leica RM2145 microtome. After deparaffinisation, antigen retrieval was performed in citrate buffer and endogenous peroxidases were blocked in H_2_O_2_. Immunochemistry staining was carried out using an avidin–biotin-based protocol (Vectastain ABC kit, Vector Laboratories, Burlingame, USA). As a primary antibody, we used an anti-CD68 antibody (diluted 1/100) (BioRad Laboratories, Hercules, CA, USA). A negative control was performed in the absence of the primary antibody. Images were acquired using an Eclipse TE300 inverted microscope (Nikon). Results were expressed as macrophages number per mm^2^ of tissue after analysis of 10–20 fields per sample.

For adipocyte size measurement, adipose tissue sections were stained with Hematoxylin QS (H-3404, Vector Laboratories) for 1 min. Adipocyte size was then determined in µm^2^ using ImageJ analysis software (http://rsbweb.nih.gov/ij/, 29 March 2021), with each cell being individually identified and measured. About 150–300 adipocytes were analysed for each sample according to Parlee et al. [35]. A frequency function was used in Excel software to assess the effects of each diet on adipocyte size repartition. For periaortic brown adipose tissue, lipid inclusions (empty fields) in the tissue were quantified using ImageJ software.

### 2.5. Statistical Analysis

Results are expressed as means ± SEM, *n* = 4–7 animals per group. The statistical analysis was performed using one-factor analysis of variance (ANOVA) and post hoc comparisons were performed with Fisher’s protected least significant difference (PLSD) test using STATVIEW software (SAS Institute Inc., Cary, NC, USA). Unless indicated, *p*-values given account for the PLSD test. Differences were considered significant when the *p*-value was under 0.05.

## 3. Results

### 3.1. Body Weight, Glucose Homeostasis and Blood Lipids

As expected, body weight and body weight gain were higher in all groups fed a high-fat diet when compared to the control group (Table 1). A non-significant increase (ANOVA: *p* = 0.11) in eWAT weight was also observed with the obesogenic diets, the trend being the strongest for the lard group and the weakest for the refined PO group.

Glucose homeostasis was assessed by measuring fasting blood glucose and insulin levels. An increase in blood glucose was observed with the crude (*p* < 0.05) and refined (*p* < 0.01) PO diets compared to controls. Fasting insulinemia increased with the three HF diets, and especially with the refined PO, almost reaching the significance level (ANOVA: *p* = 0.065). An enhancement in the HOMA-IR index was observed in all high-fat groups, and significance was only achieved with the refined PO diet (*p* < 0.01).

Blood triglycerides and total cholesterol were not significantly different between controls and HF diets groups while the highest values are observed after the refined PO diet. High-density lipoprotein (HDL) cholesterol was the lowest for the lard group (*p* < 0.05).

### 3.2. Lipid Storage in Adipose Tissues

We quantified lipid deposition in pWAT and eWAT by measuring adipocyte size (area in µm^2^) following the 12 weeks of nutritional intervention.

In eWAT, adipocyte size rose following diet supplementation with lard. The mean adipocyte surface in this group was 6829 ± 491 µm^2^ compared to 4312 ± 335 µm^2^ (*p* < 0.01) for control rats (Figure 1B). A non-significant increase in adipocyte size was also observed in rats that consumed crude (5434 ± 155 µm^2^) and refined palm oil (5461 ± 421 µm^2^) (*p* = 0.2816 and *p* = 0.1902, respectively). Concerning the adipocyte size distribution, we observed a flattening and a shift to the right of the curves following the obesogenic diets compared to the controls, and especially for the lard group (Figure 1C). This is typical of adipocyte hypertrophy in WAT. Nevertheless, distribution curves for the crude PO and refined PO groups showed an intermediate profile, whereas a second population of large adipocytes (≈8000 µm^2^) appeared to be present in the lard group.

Adipocytes were also enlarged in the pWAT of the lard group compared to control rats (3213 ± 250 μm^2^ versus 2352 ± 255 μm^2^, respectively, *p* < 0.05, Figure 2B). In contrast with eWAT, the increase in cell size in pWAT was the highest for the refined PO group (3452 ± 248 µm^2^, *p* < 0.01) with a more spread out size distribution than for other groups (Figure 2C). In addition, large (around 5000 µm^2^) adipocyte populations were observed in this group. Interestingly, the lowest adipocyte size enhancement was for the crude PO group (2905 ± 372 µm^2^, non-significant difference versus controls). For this group, the cell size distribution profile was intermediate and closer to that of control rats compared to lard and refined PO groups (Figure 2C).

Lipid storage in AoBAT was evaluated by the quantification of lipid inclusions in the tissue and expressed as the percentage represented by these areas reported for the total area of the tissue (Figure 3). A significant increase in lipid inclusions in AoBAT was observed for the refined PO (50.1 ± 2.5%, *p* < 0.01) and lard (50.2 ± 3.3%, *p* < 0.01) groups compared to controls (38.5 ± 5.2%) (Figure 3B). Interestingly, this increase was also significant (*p* < 0.01) for both when compared to the crude PO group in which lipid inclusions (39.2 ± 0.9%) were similar to control rats. In addition, dispersion of the values for lipid inclusions was low for this group in contrast to the other HF diets, as described in the box plot (Figure 3C).

### 3.3. Macrophage Recruitment in White Adipose Tissue

Macrophage number (CD68-positive cells) in eWAT was not influenced by the obesogenic diets (Figure 1D). In the control rats, we observed 6.5 ± 2.0 macrophages/mm^2^ of tissue. This number remained statistically unchanged (ANOVA: *p* = 0.6043) for the crude PO (6.8 ± 2.2 macrophages/mm^2^ of tissue) and refined PO (7.0 ± 1.5 macrophages/mm^2^ of tissue) groups and slightly lower for the lard group (4.6 ± 1.1 macrophages/mm^2^ of tissue).

Almost significantly (ANOVA: *p* = 0.0886), a tendency of macrophage infiltration in pWAT was present in the refined PO group compared to control rats (17.7 ± 3.6 vs. *7*.7 ± 1.8 macrophages/mm^2^ of tissue, respectively) (Figure 2D). Conversely, few macrophages were observed in pWAT of the crude PO and lard groups (6.9 ± 2.6 and 6.0 ± 2.4 macrophages/mm^2^ of tissue, respectively).

### 3.4. Isoprostanoid Formation in pWAT

Several isoprostanoids were studied in pWAT as markers of oxidative stress and potential biological mediators. Total isoprostanoids (free and esterified forms) were assayed for the analytes 15(*RS*)15-F_2t_-IsoP, 5(*RS*)-5-F_2t_-IsoP, 5(*RS*)-5-F_2c_-IsoP and 7(*RS*)-ST-Δ^8^-11-dihomo-IsoF (this latter molecule being an isofuran form derived from adrenic acid) [36]. Along with adipose tissue hypertrophy, lipid droplet content increased and membrane proportion decreased (as exemplified in Figure 2). This could affect isoprostane measurement considering a given weight of adipose tissue, as isoprostanoids are mainly derived from membrane phospholipids [37] and are less likely to be present within lipid droplets. Since adipocyte size was determined, we chose to express the isoprostanoid amount per adipocyte.

According to ANOVA, no significant change was observed regarding the amount of 15(*RS*)-15-F_2t_-IsoP and 5(*RS*)-5-F_2t_-IsoP in pWAT following the four different diets (Figure 4A). However, the production of 15(*RS*)-15-F_2t_-IsoP tended to be higher in the refined PO group (186 ± 17 fg/adipocyte) compared to the control (131 ± 16 fg/adipocyte) and lard (121 ± 14 fg/adipocyte) groups (ANOVA: *p* = 0.1653; Fisher’s PLSD: *p* = 0.0804 and *p* = 0.0397, respectively). Likewise, the amount of 5(*RS*)-5-F_2t_-IsoP seemed higher in the pWAT with the refined PO diet compared to the control, crude PO and lard groups, even if non-significant (ANOVA: *p* = 0.3428).

The concentration of 5(*RS*)-5-F_2c_-IsoP derived from arachidonic acid rose in the refined PO group versus controls (*p* < 0.01) and the crude PO group (*p* < 0.05) (Figure 4B). Similarly, the concentration of 7(*RS*)-ST-Δ^8^-11-dihomo-IsoF was markedly enhanced with the refined PO diet as compared to control and crude PO groups (*p* < 0.01) (Figure 4C). A non-significant increase in the amount of this oxidised lipid was also noticed with the lard group.

## 4. Discussion

Our results demonstrate that 12 weeks of palm oil-based HF diets led to specific morphologic and functional changes in pWAT compared to eWAT and AoBAT. Our main finding is that crude and refined PO induce contrasting effects on ectopic pWAT and AoBAT, although these two oils present a very similar fatty acid composition. After a refined PO diet, but not crude PO, we observed adipocyte hypertrophy, the presence of macrophages and signs of oxidative stress in pWAT.

Differences between groups in adipocytes hypertrophy and AoBAT “whitening” [38] were unlikely to be related to body weight gain, which was equivalent for all obese groups. Adipocyte hypertrophy in pWAT and lipid deposition in AoBAT were evident for lard and the refined PO groups. Conversely, for the crude PO-fed rats, hypertrophy in pWAT was moderate and AoBAT was histologically unchanged compared to control animals. We observed that eWAT mass was slightly lower, though non-significantly, in the refined PO compared to crude PO group. Less storage in eWAT could have favoured ectopic lipid deposition in pWAT and AoBAT with the refined PO diet. This is of major concern because localisation of AT is closely related to obesity-associated complications. Fat storage in subcutaneous AT rather than visceral AT limits metabolic disorders during obesity in humans [39,40] and rats [41]. In addition, preferential lipid storage in epididymal WAT could have protective effects against glucose intolerance, as reported in mice after intraperitoneal transplantation of eWAT [42].

These observations are in line with the fact that crude PO seems to have a lower impact on insulin resistance than refined PO and HOMA-IR was significantly enhanced only in the refined PO group. Nonetheless, the two PO diets affected fasting glucose, whereas no change was observed in the lard group compared to controls. This may be surprising as some previous studies concluded that there is a deleterious effect of lard on systemic and liver glucose tolerance [43]. On the other hand, the enhancement of fasting glucose observed for the crude PO group without a clear increase in HOMA-IR is quite intriguing, as insulin resistance classically precedes glucose intolerance.

We can assume that for some compounds affecting AT metabolism, the concentration changed during the PO refining process. Although fatty acid proportions were comparable for the three fats used in the present experiments, refined palm oil had a higher omega-6 to omega-3 fatty acid ratio (see Appendix A) [15]. Consumption of PUFA (polyunsaturated fatty acids) omega-3 has induced anti-inflammatory properties in AT of obese patients [44,45], and prevented or reversed insulin resistance in obese rats [46]. Contrary to omega-3, omega-6 fatty acids were unable to downregulate maternal dyslipidaemia-induced oxidative stress [47].

Another dietary factor that may have a role in the smaller alterations induced by the crude PO is its higher content in polyphenols and carotenoids compared to refined PO and lard [15,48]. Indeed, less than 1% of the carotenoid content is preserved and PO antioxidant capacity is reduced by 80% during PO refining steps (Appendix A). Increased carotenoid consumption is associated with a reduction in visceral adiposity in humans and rodents [23], while carotenoid serum and AT concentrations are inversely correlated with insulin resistance of adipose tissue and liver [24,49]. Additionally, numerous studies have shown that polyphenols such as resveratrol, curcumin or gingerenone A were able to inhibit adipogenesis and AT mass in rodent models [24,50]. Indeed, olive oil extracts rich in oleuropein, hydroxytyrosol and triterpenic acids present protective effects against adipose tissue damage and obesity-associated metabolic disorders in HF diet rodent models [51,52]. Of note, the latter studies were interested in WAT from various origins (epididymal, mesenteric, retroperitoneal, perirenal and inguinal AT) but not peripancreatic WAT. Although no direct causal effect is demonstrated, our present findings suggest that the high content in polyphenols and carotenoids of crude palm oil could have contributed to moderate adipocyte hypertrophy (or lipid inclusions for AoBAT) and, above all, to limiting inflammation and macrophage infiltration in peripancreatic WAT.

Macrophage infiltration was absent in epididymal WAT whatever the high fat diet used. In obesity, inflammation and macrophage infiltration usually develop in AT following adipocyte hypertrophy [11,53,54]. Nevertheless, the recruitment of macrophages into eWAT during a high-fat diet is not always observed [55,56], even in the presence of an overexpression of pro-inflammatory cytokines [57]. Here, we postulate that the diet (composition, duration) was not harmful enough to induce macrophage infiltration in eWAT. However, macrophage infiltration was observed in the pWAT of the refined PO group, demonstrating inflammatory adaptation in this ectopic WAT. This is consistent with the study of Laugerette and colleagues [22], in which, among four oil-enriched diets, PO induced the highest levels of IL-6 in plasma and expression of IL-1β and TLR4 in WAT. Recently, a pWAT-focused study in mice found that this AT depot is highly metabolically active and that its surgical removal worsened obesity-related hyperinsulinemia [58]. This adipose deposit appeared to play an important role in the regulation of basal and glucose-stimulated insulin levels. Interestingly, Rebuffat et al. [20,59] have reported that modified cytokine production in pWAT can affect pancreatic beta cell plasticity and function. Although cytokine production was not measured, the macrophage infiltration observed suggests that inflammation is present in peripancreatic AT with a refined PO diet. Alterations of peripancreatic WAT secretion were potentially involved in insulin secretion modifications, disruption of glucose homeostasis and the increase in HOMA-IR observed with this palm oil diet.

An in-depth characterisation of the AoBAT was not within the scope of this study but the restricted lipid deposition in AoBAT following a crude PO diet should be underlined and could have health benefits. Indeed, AoBAT is a relevant AT for cardiovascular risk in obese patients [60,61]. Some HF-based animal studies also identified that this AT as of major concern for vascular disease development [62]. It is noteworthy that Yuriko Oi-Kano et al. demonstrated in a rat HF diet model (30% palm oil) that supplementation with the olive oil phenolic extract oleuropein enhanced BAT activity, with protective effects on visceral AT [63]. Moreover, AoBAT vascularisation seems to be involved in its “whitening” and can impact whole-body glucose metabolism [38].

Inflammation, macrophage recruitment, adipocyte metabolic activity and oxidative stress generation are closely related in AT, and could increase isoprostanoid production from membrane phospholipids [37]. In the current study, isoprostanoid content was considered for one single adipocyte to limit the impact of hypertrophy (raising the proportion of lipid droplets) on the measure. A major finding was that the concentration of several isoprostanoids was increased in pWAT with the refined PO diet, indicating local oxidative stress enhancement. In addition, no significant change was observed for the crude PO group and only dihomo-isofuran concentration tended to rise with the lard diet. Isofurans, which were recently discovered [36,64], are mainly produced when oxygen tension is high and their combined measurement with isoprostanoids allows a reliable overall assessment of oxidative stress damage. Oxidative status is a finely controlled parameter in AT, regulating the proliferation and differentiation capacities of preadipocytes into mature adipocytes, and is likely involved in their dysfunction [26,65]. Oxidative stress in AT depends on the balance between ROS production (mitochondrial activity, NADPH oxidase, production by macrophages) and the activity of antioxidant defences, including superoxide dismutases or glutathione peroxidases, which are possibly reduced during obesity [66,67]. In that respect, it has been shown that a decrease in glutathione-s-transferase expression favoured a strong increase in oxidative stress in rat epididymal WAT following a cafeteria diet [68]. In our HF diet model, it could be interesting to determine if adipose tissue oxidative stress and inflammation are more related to changes in ROS production or to a drop in oxidant enzymes [27,65,69]. Oxidative stress enhancement in pWAT with a refined PO diet could be a cause and a consequence of macrophage infiltration in this AT. Indeed, ROS production by adipocytes promotes macrophage recruitment, and the latter are potential sources of ROS [25,26,66]. More than just oxidative stress markers, isoprostanoids are bioactive compounds and 15-F_2t_-IsoP functional effects have been widely studied. Remarkably, this isoprostanoid can inhibit angiogenesis [30] or stimulate macrophage migration and their cytokine production [31]. These properties could have participated in pWAT dysfunction in the refined PO group. In addition, our group recently observed that several isoprostanoids, including 15-F_2t_-IsoP and its epimer, inhibit insulin secretion in isolated islets (unpublished data) [70]. This could have favoured the glucose homeostasis disruption observed with the refined PO through local biocommunication between pWAT and the pancreas. Regarding 5(*RS*)-5-F_2c_-IsoP and 7(*RS*)-ST-Δ^8^-11-dihomo-IsoF, a study of their biological effects received little attention but their high concentration in pWAT following a refined PO diet may have had functional consequences.

## 5. Conclusions

To conclude, a high dietary intake of refined PO led to an increase in lipid inclusions in periaortic BAT and adipocyte hypertrophy in peripancreatic WAT. The latter displayed signs of inflammation and increased oxidative stress. Dysfunction of this adipose tissue could have influenced pancreas and glucose homeostasis by the production of pro-inflammatory adipokines or lipid mediators according to local biocommunication. Our observations were related to a high intake of refined PO and adverse effects would be unlikely during normal consumption of this oil as part of a healthy diet [4]. Moderate ectopic lipid accumulation was observed with the crude PO diet and no macrophage infiltration or oxidative stress was detected in peripancreatic WAT. It is suggested that the high antioxidant content of crude PO may alleviate its deleterious effects on this adipose deposit. This study shows that not only composition in fatty acids, but mostly the amount of valuable natural compounds as antioxidants of oils used in high-fat diets, can have morphological and functional consequences for adipose tissues. Our results also show pWAT-specific adaptations to PO-rich high-fat diets.

## Figures and Tables

**Figure 1 nutrients-13-01134-f001:**
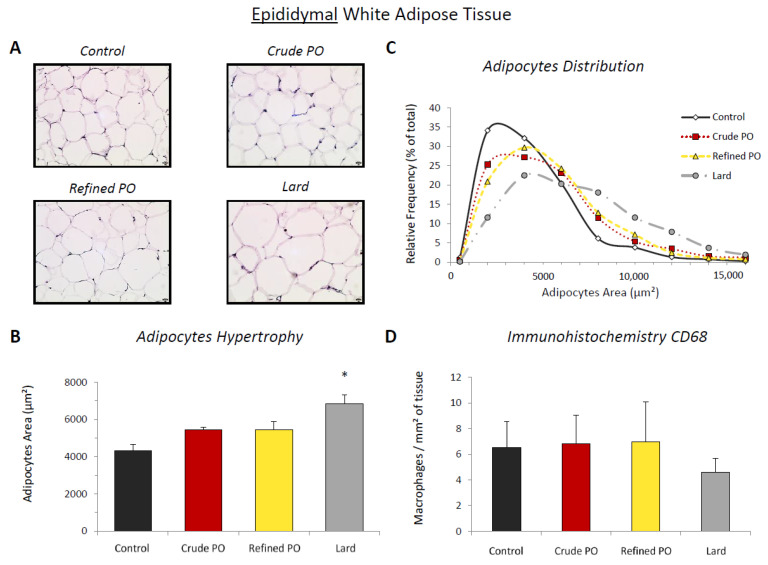
Adipocyte size and macrophage infiltration in epididymal white adipose tissue (eWAT). (**A**) Representative microphotographs of haematoxylin staining (×200; scale bar = 20 µm); (**B**) quantitative analysis of adipocyte area (µm^2^); (**C**) relative frequency of adipocyte area (%); (**D**) quantitative analysis of immunohistochemistry for CD68 (number of positive cells/mm^2^ of adipose tissue). Data are shown as means ± SEM, *n* = 4–7 animals per group. * *p* < 0.05 vs. control diet.

**Figure 2 nutrients-13-01134-f002:**
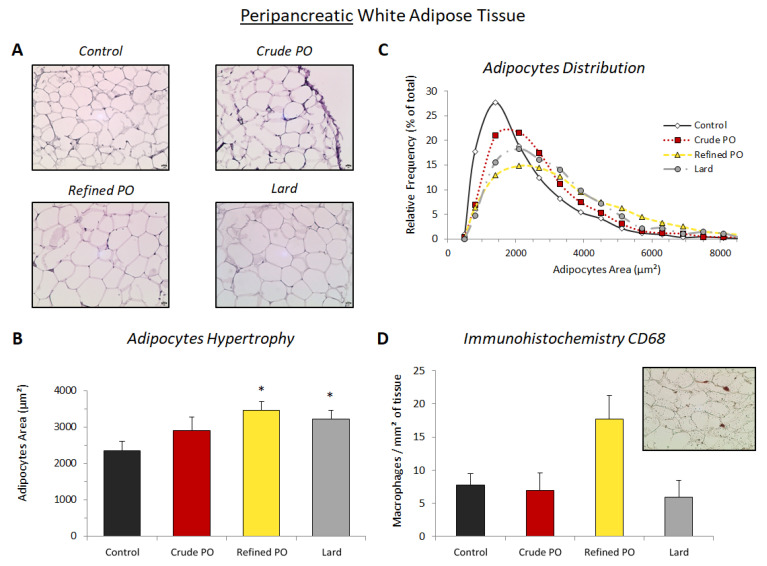
Adipocyte size and macrophage infiltration in peripancreatic adipose tissue (pWAT). (**A**) Representative microphotographs of haematoxylin staining (×200; scale bar = 20µm); (**B**) quantitative analysis of adipocyte area (µm^2^); (**C**) relative frequency of adipocyte area (%); (**D**) quantitative analysis of immunohistochemistry for CD68 (number of positive cells/mm^2^ of adipose tissue). Inset presents an example of CD68 staining in refined PO. Data are shown as means ± SEM, *n* = 4–7 animals per group. * *p* < 0.05 vs. control diet.

**Figure 3 nutrients-13-01134-f003:**
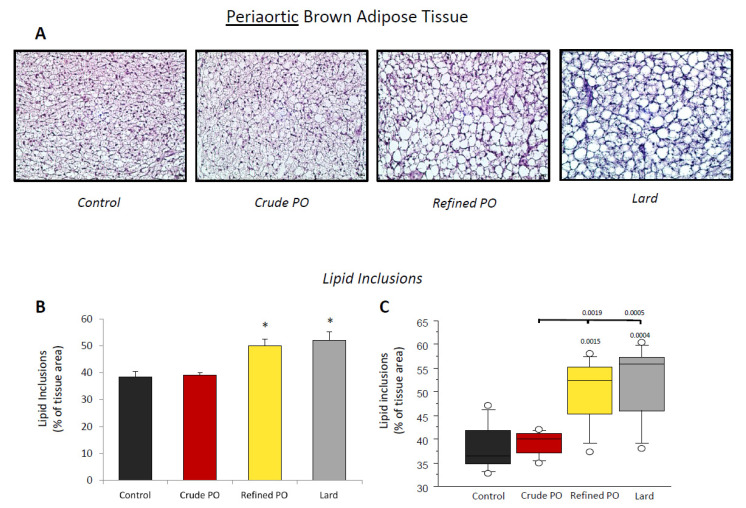
Lipid inclusions in periaortic brown adipose tissue (AoBAT). (**A**) Representative microphotographs of haematoxylin staining (×200; scale bar = 20 µm); (**B**) quantitative analysis of lipid inclusions as % of total tissue area; (**C**) box plot displaying the distribution of lipid inclusions. Quantification of lipid inclusion was realised in 10–30 fields per rat. Data are shown as means ± SEM, *n* = 7 animals per group. * *p* < 0.05 vs. control diet.

**Figure 4 nutrients-13-01134-f004:**
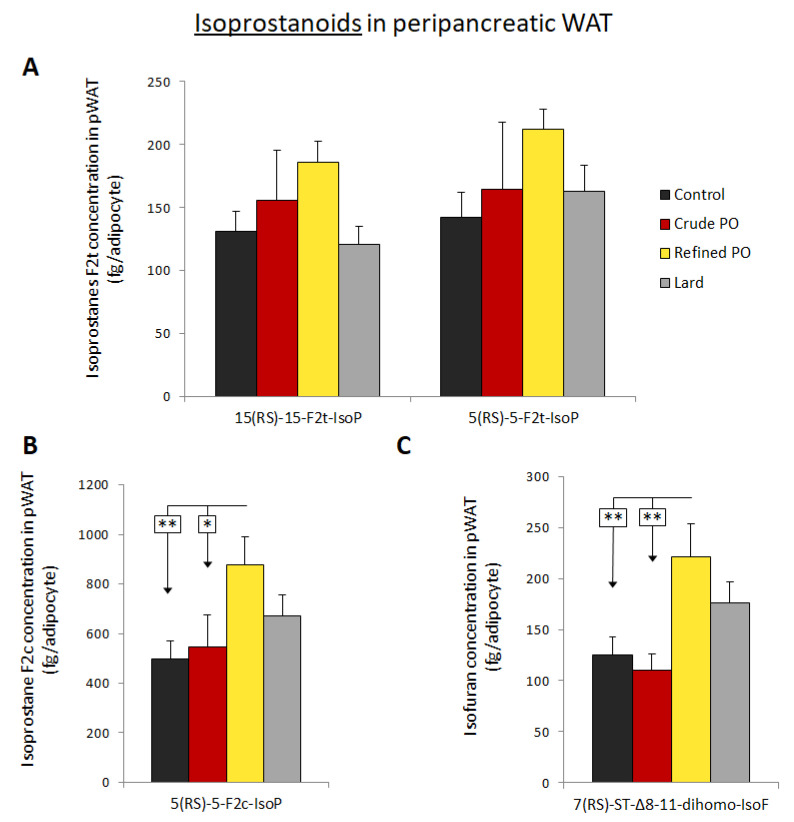
Impact of high intake of palm oils and lard on isoprostanoid concentration in peripancreatic adipose tissue (pWAT). (**A**) Quantification of isoprostanoids with a trans configuration: 15*(RS)*-15-F2t-IsoP and 5*(RS)*-5-F2t-IsoP (fg/adipocyte). (**B**) Quantification of 5*(RS)*-5-F2c-IsoP, an isoprostanoid with a cis configuration (fg/adipocyte). (**C**) Quantification of the isofuran 7*(RS)*-ST- Δ^8^-11-dihomo-IsoF (fg/adipocyte). Isoprostanoids were assayed from 200 mg of adipose tissue per rat. Data are shown as means ± SEM, *n* = 6–7 animals per group. * *p* < 0.05 or ** *p* < 0.01 vs. control diet.

**Table 1 nutrients-13-01134-t001:** Body weight, glucose homeostasis and blood lipids.

	Control	Crude PO	Refined PO	Lard	*p*-Value
Initial body weight (g)	209 ± 4	203 ± 1	204 ± 2	208 ± 3	NS
Final body weight (g)	527 ± 11	609 ± 14 **	612 ± 20 **	611 ± 23 **	0.0065
Body weight gain (g)	318 ± 9	406 ± 14 **	408 ± 19 **	403 ± 22 **	0.0022
eWAT weight (g)	12.6 ± 1.8	20.1 ± 2.1	16.6 ± 3.5	21.4 ± 2.9	NS
Fasting glucose (mg/dL)	136.0 ± 3.5	155.0 ± 5.8 *	157.1 ± 3.7 **	131.6 ± 5.7	0.0015
Fasting insulin (mU/L)	53.5 ± 7.6	108.6 ± 22.9	147.0 ± 29.9	93.6 ± 23.6	NS
HOMA-IR	18.1 ± 2.8	41.7 ± 9.1	57.6 ± 12.0 **	31.6 ± 8.7	0.0344
Blood TG (mM)	1.03 ± 0.15	0.99 ± 0.11	1.33 ± 0.21	1.01 ± 0.15	NS
Total cholesterol (mM)	1.97 ± 0.16	1.75 ± 0.15	2.00 ± 0.14	1.55 ± 0.09	NS
HDL cholesterol (mM)	1.48 ± 0.12	1.34 ± 0.85	1.56 ± 0.12	1.06 ± 0.12 *	0.0179

Results are expressed as mean ± SEM, *n* = 7 animals per group. After one-way ANOVA (*p*-value indicated), data were compared by a Fishers’ protected least significant difference (PLSD) test: * *p* < 0.05 and ** *p* < 0.01 vs. control group; PO: palm oil, NS: non-significant, eWAT: epididymal white adipose tissue, HOMA-IR: homeostasis model assessment of insulin resistance, TG: triglycerides, HDL: high-density lipoprotein.

## Data Availability

The datasets generated and analysed during the current study are available from the corresponding author on reasonable request.

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
