# Peer review of "Peripancreatic Adipose Tissue Remodeling and Inflammation during High Fat Intake of Palm Oils or Lard in Rats"

_nutrients, 2021, doi:10.3390/nu13041134_

Round 1

Reviewer 1 Report

The manuscript is well-written and well-organized.

Despite a  wide range of references, I recommend inlcuding the reports on extra virgin olive oil, e.g. Critical Reviews in Food Science and Nutrition, 2020, doi: 10.1080/10408398.2020.1799930; Nutrients, 2020, 12(2),323; Journal of Nutritional Biochemistry, 2017, 40, pp. 209-218; Food and Chemical Toxicology, 2017, 107, pp. 329-338 in the Discussion section.

Author Response

Dear reviewer 1,

Thank you for your comments. Please find enclose my replies to your observations.

Best regards,

JL

Reviewer 2 Report

The study "Remodeling and Inflammation of Peripancreatic Adipose Tissue during High Fat Intake of Palm Oils or Lard in Rats" by Laget et al. has been reviewed. In general, a diet rich in fatty acids contributes to obesity and various health issues. Palm oil is a rich source of fatty acids and consuming access causes lipid accumulations in the adipose tissues. The authors discovered that consuming crude palm oil resulted in less lipid deposit than consuming refined palm oil or lard.  According to the authors, the antioxidants found in crude palm oil, may have protected adipose tissue.

The findings are agreeable and well-explained by a thorough literature review.

Author Response

Dear reviewer 2,

Thank you for your comments. Please find enclosed my replies to your observations.

Best regards,

JL

Reviewer 3 Report

In this study, Laget et al. investigated the respective effects of intake of high amounts of palm oils and lard. While the authors’ findings are potentially of interest, there are major concerns in experimental design and thereby the authors’ conclusions. First, there was no definition of the ‘high intake of palm oils and lard’. A corresponding group of ‘low’ intake of palm oils and lard should be included for comparisons. The authors only used male rats; therefore, the findings in Table 1 may not be significant in the female rats. Moreover, palm oils have no effects or have been regarded as beneficial for diabetics, including humans. Thus, the authors’ findings appear to be contradictory to many others’ reports. Second, the positive findings in in Table 1 was of marginal significance. For instance, the normal levels of fasting glucose in rats are below 135±15 mg/dL. Less than 200 mg/dl at most confers low risk of diabetes. The analysis of adipose composition in Figures 1 and 2 was not significant. In Figure 1D and 2D, the macrophage infiltration in the fat also showed no significant inflammation. Moreover, the lard group even showed reduced inflammation. Third, the authors should investigate the blood lipid profiles (total cholesterol, low-density lipoprotein cholesterol, high-density lipoprotein cholesterol, triacylglycerol) and measure BMI, and include these data in Figure 3 to draw conclusions. Fourth, the authors showed the levels of isoprostanoids in fat tissues as a readout of oxidative stress. However, what is the physiological meaning of increased oxidative stress in the fat as opposed to the findings in Figures 2 and 3? Evidence of oxidative stress at the cellular level, e.g. IHC or western blotting, should be provided.

Author Response

Dear reviewer 3,

Thank you for your comments. Please find enclose my replies to your observations.

Best regards,

JL

Reviewer 4 Report

Concerns:

  1. Introduction: It would be beneficial to present more information in the rationale why authors were interested in peripancreatic adipose tissue remodeling induced by refined palm-oil based HFD in the view of inflammation and development of insulin resistance.
  2. Results: Authors need to measure cytokines production by peripancreatic adipose tissue as macrophages infiltration is
  3. Figure legends: Number of animals needs to be specified for each experiment.
  4. Discussion: in the view of their and Chanclon et al study (ref#56), authors need to discuss possible consequences of altered inflammation in pWAT on insulin secretion.

Author Response

Dear reviewer 4,

Thank you for your comments. Please find enclose my replies to your observations.

Best regards,

JL

Round 2

Reviewer 4 Report

  1. Lane 168: “Results were expressed as means ±SEM, n=7 animals per group”- this needs to be changed in the view of actual number of mice per group in every set of data.

Author Response

Response to reviewer 4 comments:  
  1. Lane 168: “Results were expressed as means ±SEM, n=7 animals per group”- this needs to be changed in the view of actual number of mice per group in every set of data.

Thank you for your comment.

Line 168: The sentence have been modified to "Results were expressed as means ±SEM, n=4-7 animals per group."

Jonas Laget